# The amino acid composition of a protein influences its expression

Reece Thompson, Benjamin Simon Pickard*

Strathclyde Institute of Pharmacy and Biomedical Science, University of Strathclyde, Glasgow, United Kingdom

* benjamin.pickard@strath.ac.uk

**Data Availability Statement:** All relevant data are within the paper and its Supporting Information files.

**Funding:** The author(s) received no specific funding for this work.

## Abstract

The quantity of each protein in a cell only is only partially correlated with its gene transcription rate. Independent influences on protein synthesis levels include mRNA sequence motifs, amino acyl-tRNA synthesis levels, elongation factor action, and protein susceptibility to degradation. Here we report that the amino acid composition of a protein can also influence its expression level in two distinct ways. The nutritional classification of amino acids in animals reflects their potential for scarcity–essential amino acids (EAA) are reliant on dietary supply, non-essential amino acids (NEAA) from internal biosynthesis, and conditionally essential amino acids (CEAA) from both. Accessing public proteomic datasets, we demonstrate that a protein's CEAA sequence composition is inversely correlated with expression– a correlation enhanced during rapid cellular proliferation–suggesting CEAA availability can limit translation. Similarly, proteins with the most extreme compositions of EAA are generally reduced in abundance. These latter proteins participate in biological systems such as taste and food-seeking behaviour, oxidative phosphorylation, and chemokine function, and so linking their expression to EAA availability may act as a homeostatic response to malnutrition. Protein composition can also influence general human phenotypes and disease susceptibility: stature proteins are enriched in CEAAs, and a curated dataset of over 700 cancer proteins is significantly under-represented in EAAs. We also show that individual amino acids can influence protein expression across all kingdoms of life and that this effect appears to be rooted in the unchanging structural and mRNA encoding features of each amino acid. Species-specific environmental survival pathways are shown to be enriched in proteins with individual amino acid compositions favouring higher expression. These two forms of amino acid-driven protein expression regulation promise new insights into systems biology, evolutionary studies, experimental research design, and public health intervention.

## Introduction

The regulated transcription of mRNA from DNA, and subsequent translation into effector proteins, underlies all of life's dynamic processes. However, a typical gene's levels of mRNA and protein only show a correlation of 0.6 [1–3], indicating the presence of DNA-independent

**Competing interests:** The authors have declared that no competing interests exist.

regulatory influences on translation. Those influences are complex and incompletely understood [4–6] but include mRNA sequence motifs, compatibility between mRNA codon choice and corresponding tRNA-amino acid availability [7, 8] and the complex regulation of translation initiation, elongation, termination, and protein degradation.

Certain amino acid concentration changes are known to be detected by the mTORC1 signalling pathway which elicits molecular and cellular changes according to nutritional state [9]. However, the direct impact of global amino acid scarcity on protein translation is underexplored, despite supply characteristics defining an important amino acid classification system in animals [10]. That classification comprises *essential* amino acids (EAA) required from diet, *non-essential* amino acids (NEAA) obtained through biosynthesis, and an ill-defined intermediate class, *conditionally essential* amino acids (CEAA), requiring supplementation from diet during development and periods of stress or illness [11, 12]. Over 500 million years ago the new animal kingdom was, in part, distinguished by a coordinated inactivation of biosynthetic pathways for the EAA class [13–15]. The resulting switch from an autotrophic (nutrient synthesising) to auxotrophic (nutrient requiring) lifeway obliged animals to obtain EAAs from a diet of plants, bacteria, or, indirectly from prey that fed on those sources. The opportunity for increased biological size and complexity offered by the energetic efficiency of a higher trophic level has been of demonstrable advantage to animals but it created vulnerability to situational deficits in dietary supply of EAA and possibly CEAA. Such deficits would likely limit tRNA-amino acid synthesis and availability, slowing protein translation rate, and decreasing expression–with inevitable phenotypic consequences.

Here we explore that possibility, and present supportive evidence obtained from quantitative proteomics datasets that a protein's amino acid composition correlates with its expression in two distinct ways. Firstly, we show that extreme proportions of EAA and CEAA nutritional class amino acids in an animal protein exert a negative influence on expression, suggesting that amino acid demand can outstrip supply during translation. Secondly, we show that the proportions of individual amino acids in a protein influence its expression and that this effect is shared across all kingdoms of life - and derived from the unchanging structural and encoding features of amino acids. We propose that evolution has harnessed amino acid effects on expression to select protein compositions that confer advantageous responses during environmental stress.

## Results

### Effects of nutritional amino acid classes on protein expression

We first examined how the amino acid nutritional class composition of every human protein influences its expression. As a start, the mass spectrometry-derived expression levels of 9,399 liver proteins were accessed from the public proteomics repository, PaxDB [16] (**Methods**). **Fig 1A** shows proteins ranked from low-to-high frequency of each of the three nutritional classes plotted against a conservative moving median protein expression level. For most proteins, a greater compositional frequency of CEAA (fCEAA) was generally associated with a modest decrease in protein expression, whereas greater EAA (fEAA) was associated with a modest increased expression. However, at the extremes of composition, the outcome was more striking with very high fEAA and fCEAA both repressing expression, and very low fEAA and fCEAA permitting higher expression. At these extremes, we interpret the apparent fNEAA influence on expression as just a passive numerical consequence of active fCEAA or fEAA effects. These findings are striking in two regards: they are naïve of mRNA expression information, and data are derived from human donors without known dietary amino acid deficiency. We propose an explanation in which inadequate nutritional supply (EAA) or

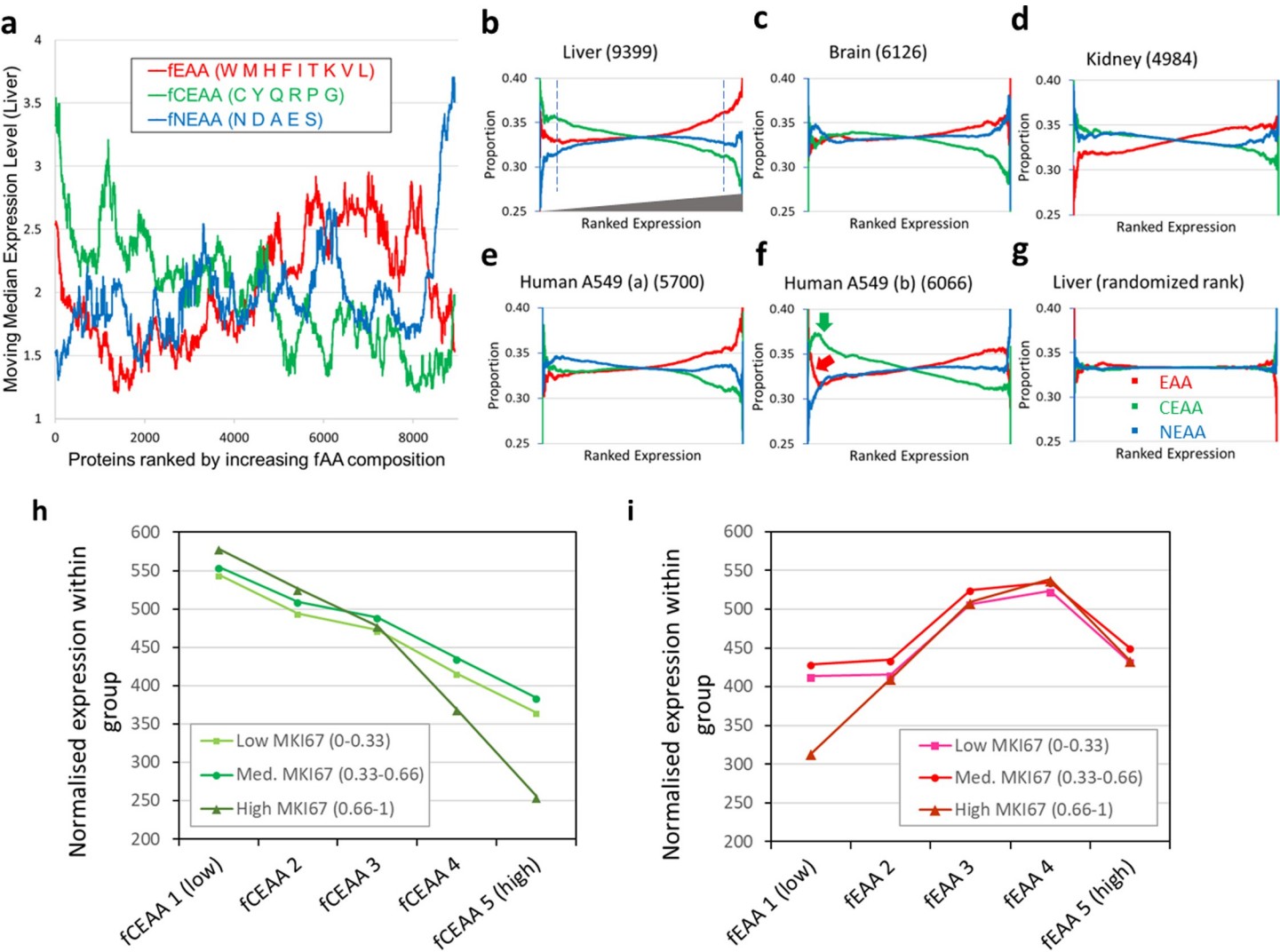

**Fig 1. The relative proportions of the three nutritional amino acid classes within a protein change with expression level - an effect enhanced by proliferation rate.**
(**a**) Proteins were ranked by frequency of each of the three nutritional classes (fNEAA, blue; fCEAA, green; fEAA, red. Single-letter amino acid codes are shown) and plotted against a moving median of liver expression levels (periodicity = 5% of total protein number) to determine composition influence on expression. (**b-g**) A smoothing procedure (see **Methods** and **S1 File**) was applied to visualise trends in relative, ranked amino acid class proportion when plotted against increasing ranked protein expression levels in human liver, brain, and kidney, and in A549 lung cancer cells lines with different proliferation rates (data from PaxDB). The full set of human organ data is presented in **S1 Fig**. In (**b**), dashed lines have been added to highlight changes in proportions of the amino acid classes between low- and high-expression proteins. A visual indication of increasing expression rank is shown. In (**f**), green (CEAA) and red (EAA) arrows have been added to indicate the changes in proportion from (**e**). In (**g**), liver proteins have been randomised with respect to expression rank, removing trends in amino acid class representation and confirming the validity of the smoothing approach. A colour legend has also been added. (**h/i**) The impact of cellular proliferation rate on protein expression was examined in PaxDB data from 26 cell lines from multiple laboratories stratified into three cohorts (low, medium, and high proliferation) according to rescaled/normalised expression level of the proliferation marker MKI67 (Ki67) (see **Methods**). Proteins were additionally subdivided into 5 amino acid class frequency ranges (fEAA/fCEAA 1–5, with 5 having the greatest representation of the amino acid class). Total protein expression levels were calculated within each of the 15 groups and plotted.

biosynthesis (CEAA) particularly constrains the expression of proteins with extreme EAA/ CEAA compositions.

An alternative approach to data visualisation, focusing on relative amino acid compositions of each protein across the range of expression, was applied to data from eight human organs and a lung alveolar basal epithelial adenocarcinoma cell line, A549. The accessed organ expression datasets (PaxDB) had already been integrated from several analyses and normalised at

source. Ranked protein expression levels were plotted against smoothed relative EAA:CEAA: NEAA proportions for each protein (**Methods**, **Figs 1B–1G,** and **S1**). The right side of every component image is largely conserved in appearance: those proteins with the highest expression have amino acid compositions that avoid extremes, tending to the whole-proteome mean frequencies (fEAA, 0.41; fCEAA, 0.28; fNEAA, 0.31). By contrast, the left side of each image was observed to fall into one of three distinct profiles. In the first, **Figs 1B** and **S1** liver, heart, and male and female gonads show a marked increase in proteins with high CEAA proportion at lower expression levels (green arrow). One interpretation of this profile is that a biosynthetic shortfall of CEAAs in these tissues results in reduced expression of CEAA-rich proteins, clustering them to the left of the diagram. In **Figs 1C** and **S1** pancreas and brain tissues do not show such trends suggesting these tissues may have different steady-state supplies or synthesis of amino acids. In **Figs 1D** and **S1** kidney and lung tissues show a greater proportion of EAA/ NEAA amino acids in proteins with low expression. These three profile types may reflect inherent organ features and states such as non-proteogenic amino acid use (gluconeogenesis), local proliferation rate, basal metabolic rate, or amino acid transport capacities. In this way, organ-specific categories of 'essential' and 'conditionally essential' amino acids may practically differ from the whole organism definition.

Of note, these observations suggest that most proteins have arrived at an evolutionary balance between the selection of appropriate amino acids (to meet requirements of structure/ functionality), and the selection of amino acid compositions that avoid supply constraints on expression. Aside from hydrophobic amino acids typically being EAAs, there is relatively little alignment between nutritional amino acid classes and the functional amino acid classifications (polar/acidic/basic, etc.) suggesting a degree of flexibility in amino acid selection.

In **Fig 1E and 1F**, two markedly different profiles are shown for the same lung cancer cell line, A549. We hypothesised that differing tissue culture protocols in the source laboratories affected amino acid availability. In **Fig 1E** (A549 a, data from [17], #id: 3312331274) there is little evidence for constrained protein expression. However, in **Fig 1F** (A549 b, data from [18], #id: 878737823) - with cells described as actively proliferating - a substantial increase in CEAA and EAA proportions (green and red arrows) was observed at lower expression levels, perhaps indicating that proteins with greater proportions of those two amino acid classes were inefficiently translated. A suspected interaction between proliferation rate, amino acid nutritional class and expression was therefore examined in PaxDB expression data from 26 cell lines (listed in **Methods**). Wide provenance, intrinsic cell line expression differences, and uncertain culture conditions at the time of protein isolation required an objective means to stratify cell line data by proliferation rate. A normalised protein expression level was derived for established proliferation marker Ki67 (MKI67) within each cell line (**Methods**). Cell lines were stratified into 3 groups by this proliferation rate proxy, as well as into 5 fEAA or fCEAA composition classes. When total protein expression levels within each of the resulting fifteen subdivisions were summed and plotted (**Fig 1H and 1I**), we saw evidence for expression influenced in two ways that are consistent with the organ analyses. Firstly, a proportionately negative influence on expression exists across the full range of fCEAA, which is further intensified (green arrow) during rapid proliferation (**Fig 1H**). We speculate that proliferation is accompanied by a substantial demand for new protein synthesis in daughter cells and that conditionally essential amino acid biosynthetic pathways are, by definition, unable to meet this situational demand, leading to a shortfall of these amino acids and a consequent impact on protein synthesis of CEAA-rich proteins. Secondly, a modest negative effect of on expression was observed only for those proteins with the very highest fEAA (perhaps determined by the limits of EAA availability in media and their cellular uptake) (**Fig 1I**).

## Individual amino acid effects on protein expression

We looked beyond nutritional supply constraint effects on amino acid groupings to determine if fundamental effects on protein expression existed at the individual amino acid level. Multiple Linear Regression (MLR) was used to determine individual amino acid frequency effects on expression within the same eight human organs and, in parallel, the root of plant *Arabidopsis thaliana*, the fungus *Saccharomyces cerevisiae*, and bacterium *Escherichia coli* (**Fig 2 and S1 File**). The individual amino acid effects on expression were substantially conserved in scale and direction between human organs and across species. Increased representation of amino acids lysine (K), glycine (G), alanine (A), and valine (V) largely correlated with increased expression of a protein, whereas tryptophan (W), arginine (R), cysteine (C), and serine (S), largely correlated with reduced expression. Human-specific inhibitory effects were seen for isoleucine (I) and proline (P), and methionine (M) was consistently inhibitory in non-animal species. Aspartate (D) only showed a positive influence in humans. These MLR models are statistically highly significant, but only generate modest $r^2$ values within a range of 0.02 to 0.13 for global prediction of protein expression in the absence of mRNA expression data or nutritional deficiency. We noted that the largest $r^2$ values were observed for human ovary, heart, and testis, all represented within the subgroup represented in **Figs 1B** and **S1A**, and for the two microorganisms cultured in proliferation-driving conditions. This suggested that there may be overlapping effects between the two models of amino acid effect on expression described here. This finds further support in the over-representation of CEAAs within the group of amino acids showing MLR negative effects. Surprisingly, serine (a NEAA not expected to be limiting) was also negatively associated with expression level. Serine's vital role as a carbon source for the synthesis of other amino acids, and the pathological consequences of its deficiency in humans has prompted a recent call for its reclassification as a CEAA [19].

The pan-species nature of these amino acid effects was effectively demonstrated by testing the ability of the *E. coli* MLR model from **Fig 2** and **S1 File** to predict trends in actual global human protein expression (**Fig 3**). Firstly, the original human liver MLR model prediction for each protein's expression level was plotted against that protein's true expression and a moving average trendline fitted (**Fig 3A**). From lowest to highest 'window' of prediction, the trendline described three orders of expression magnitude, as might be expected given this was the source data for the MLR model. However, applying the bacterial MLR model to human liver

| | EAA | CEAA | CEAA | NEAA | EAA | CEAA | CEAA | EAA | EAA | EAA | NEAA | EAA | NEAA | CEAA | EAA | NEAA | EAA | NEAA | CEAA | EAA | | |
|---|---|---|---|---|---|---|---|---|---|---|---|---|---|---|---|---|---|---|---|---|---|---|
| | **W** | **R** | **C** | **S** | **I** | **P** | **Q** | **M** | **L** | **H** | **N** | **F** | **E** | **Y** | **T** | **D** | **V** | **A** | **G** | **K** | Significance (F) | Adjusted R Square |
| *E. coli* | -0.08 | -0.04 | -0.17 | -0.05 | 0.00 | 0.00 | 0.00 | -0.05 | -0.05 | 0.00 | 0.00 | 0.00 | 0.10 | -0.07 | 0.00 | 0.00 | 0.08 | 0.07 | 0.06 | 0.17 | 3.93E-117 | 0.13 |
| *S. cerevisiae* | 0.00 | 0.05 | -0.08 | -0.04 | 0.00 | 0.00 | 0.00 | -0.14 | 0.00 | 0.00 | 0.00 | 0.00 | 0.00 | 0.00 | 0.00 | 0.00 | 0.22 | 0.22 | 0.11 | 0.14 | 4.19E-130 | 0.11 |
| *A. thaliana* Root | -0.08 | -0.10 | -0.10 | -0.09 | 0.00 | 0.00 | -0.05 | -0.08 | -0.07 | 0.00 | -0.06 | 0.00 | 0.03 | 0.00 | 0.07 | 0.00 | 0.06 | 0.10 | 0.08 | 0.06 | 3.09E-127 | 0.06 |
| *H. sapiens* Ovary | -0.14 | -0.05 | -0.09 | -0.09 | -0.08 | -0.05 | -0.05 | 0.00 | -0.03 | 0.00 | 0.00 | 0.00 | -0.04 | 0.00 | 0.00 | 0.10 | 0.06 | 0.06 | 0.05 | 0.12 | 1.98E-120 | 0.08 |
| *H. sapiens* Testis | -0.14 | -0.04 | -0.04 | -0.07 | 0.00 | -0.03 | 0.00 | 0.00 | 0.00 | 0.00 | 0.00 | 0.00 | 0.00 | 0.09 | 0.05 | 0.14 | 0.09 | 0.09 | 0.10 | 0.14 | 6.32E-109 | 0.06 |
| *H. sapiens* Heart | -0.11 | 0.00 | -0.05 | -0.06 | 0.00 | -0.05 | 0.00 | 0.10 | 0.00 | 0.00 | 0.00 | 0.00 | 0.00 | 0.07 | 0.12 | 0.07 | 0.06 | 0.12 | 0.09 | 0.10 | 7.57E-80 | 0.06 |
| *H. sapiens* Pancreas | -0.16 | -0.05 | -0.07 | -0.08 | 0.00 | -0.09 | 0.00 | -0.14 | -0.07 | -0.11 | 0.00 | 0.00 | -0.04 | 0.00 | 0.00 | 0.05 | 0.00 | 0.00 | 0.10 | 0.05 | 7.86E-57 | 0.05 |
| *H. sapiens* Brain | -0.22 | -0.08 | -0.05 | 0.00 | -0.08 | -0.08 | 0.00 | 0.00 | 0.00 | 0.00 | 0.00 | 0.00 | 0.00 | 0.09 | 0.06 | 0.10 | 0.00 | 0.06 | 0.13 | 0.04 | 3.42E-42 | 0.04 |
| *H. sapiens* Kidney | -0.24 | -0.17 | 0.00 | 0.00 | -0.20 | -0.12 | 0.00 | 0.00 | 0.00 | 0.00 | 0.00 | 0.00 | 0.00 | 0.00 | 0.00 | 0.00 | 0.00 | 0.00 | 0.19 | 0.09 | 1.88E-25 | 0.03 |
| *H. sapiens* Liver | -0.13 | -0.09 | 0.00 | -0.06 | 0.00 | -0.04 | -0.08 | 0.00 | 0.00 | 0.00 | 0.00 | 0.00 | 0.00 | 0.00 | 0.07 | 0.00 | 0.17 | 0.10 | 0.12 | 0.14 | 1.50E-40 | 0.02 |
| *H. sapiens* Lung | -0.21 | -0.15 | 0.00 | -0.10 | -0.27 | -0.12 | -0.16 | 0.00 | 0.00 | 0.00 | 0.00 | 0.00 | 0.00 | 0.00 | 0.00 | 0.00 | 0.00 | 0.00 | 0.00 | 0.00 | 4.90E-15 | 0.02 |
| Average | -0.14 | -0.07 | -0.06 | -0.06 | -0.06 | -0.05 | -0.03 | -0.03 | -0.02 | -0.01 | -0.01 | 0.00 | 0.00 | 0.02 | 0.03 | 0.04 | 0.07 | 0.07 | 0.09 | 0.10 | | |

**Fig 2. Pan-species conservation of individual amino acid influences on protein expression levels.** Human organ data are shown in the lower section and species data in the upper section. Numbers in cells represent the normalised magnitudes for statistically significant multiple linear regression (MLR) coefficients for each amino acid in each species or organ. Amino acids have been ordered left-to-right from greatest average negative effect (shades of red) to greatest average positive effect (shades of green) on protein expression. Human amino acid nutritional class assignments are shown at the top. Adjusted correlation values and statistical significance are shown on the extreme right for each of the 11 MLRs. See **S1 File** for full MLR data.

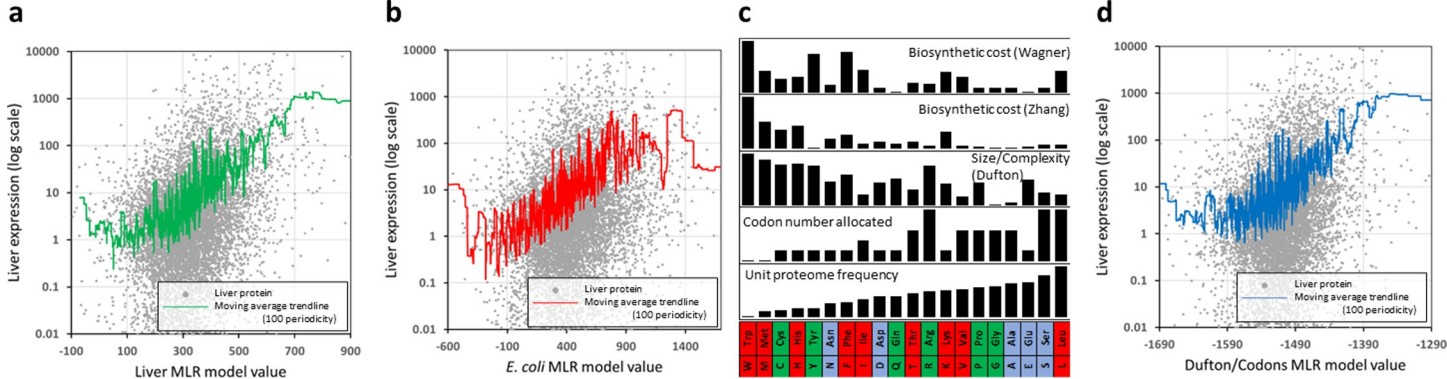

**Fig 3. Pan-species effects of amino acids on protein expression can be largely explained by two fundamental parameters.** The two expression prediction models generated by MLR analysis of individual amino acid effects on human liver and E. coli protein expression levels (detailed in **Fig 2** and **S1 File**) were tested for their ability to predict global liver protein expression. Each individual protein is shown as a green dot representing model predicted expression and actual human liver expression (log scale). Model ability is visualised by plots of moving averages (green, red, blue: periodicity of 100 proteins). (**a**) Liver model on liver expression analysis. (**b**) E. coli model on liver expression analysis. (**c**) Individual properties of amino acids (identified by their the one- and three-letter designations at the bottom) are visualised along with their amino acid frequency in the unit human proteome, and their nutritional class (red = EAA, green = CEAA, blue = NEAA). (**d**) A model combining Dufton score and number of codons allocated to each amino acid was tested for its ability to predict liver protein expression levels.

expression still successfully predicted a range of moving average liver expression spanning two orders of magnitude (**Fig 3B**).

To explain these universal effects of amino acids on protein expression we considered three fundamental amino acid properties as potential influences. The first property examined - the number of synonymous codons assigned to each amino acid - stemmed from knowledge of the importance of both aminoacyl-tRNA availability and codon preference for efficient translation. Secondly, three related models of amino acid biosynthetic cost were applied to determine if metabolic economisation pressures ('thriftiness') over the course of animal evolution had selected for protein compositions biased towards the 'cheapest to make' amino acids. In the Akashi [20] and Wagner models [21], the cost of synthesis for each amino acid is measured as high-energy phosphate bond equivalents and, in the Zhang model [22], this is further refined by including amino acid degradation constants. The third assessed property was the composite 'Dufton score' [23], assigned to amino acids based on their spatial volume and chemical complexity - encapsulating the biosynthetic, structural, and functional parameters of each amino acid. **Fig 3C** illustrates the relative magnitudes of these properties for each amino acid (tabulated in full in **S1 File**). A MLR analysis of all properties applied to the human liver expression data indicated significant contributions from the number of codons allocated ($p = 5.2 \times 10^{-14}$) and the Dufton score ($p = 5.2 \times 10^{-13}$), but no significant influence of energetic cost. Combined in a single model, these two simple and immutable amino acid properties were sufficient to generate a moving average trendline describing almost three orders of magnitude of average liver expression (**Fig 3D**).

## Protective biological systems at extremes of amino acid composition

A prediction from our findings is that animal proteins with extreme EAA or CEAA amino acid compositions would be the most sensitive to states of amino acid deficiency or physiological stress–with this sensitivity manifest as reduced expression. We theorised that these proteins might have evolved these counterintuitively extreme compositions and expression properties as part of beneficial homeostatic responses to nutritional and physiological adversity. The fEAA and fCEAA composition of all 20,397 human proteins was visualised (**Fig 4A**). We

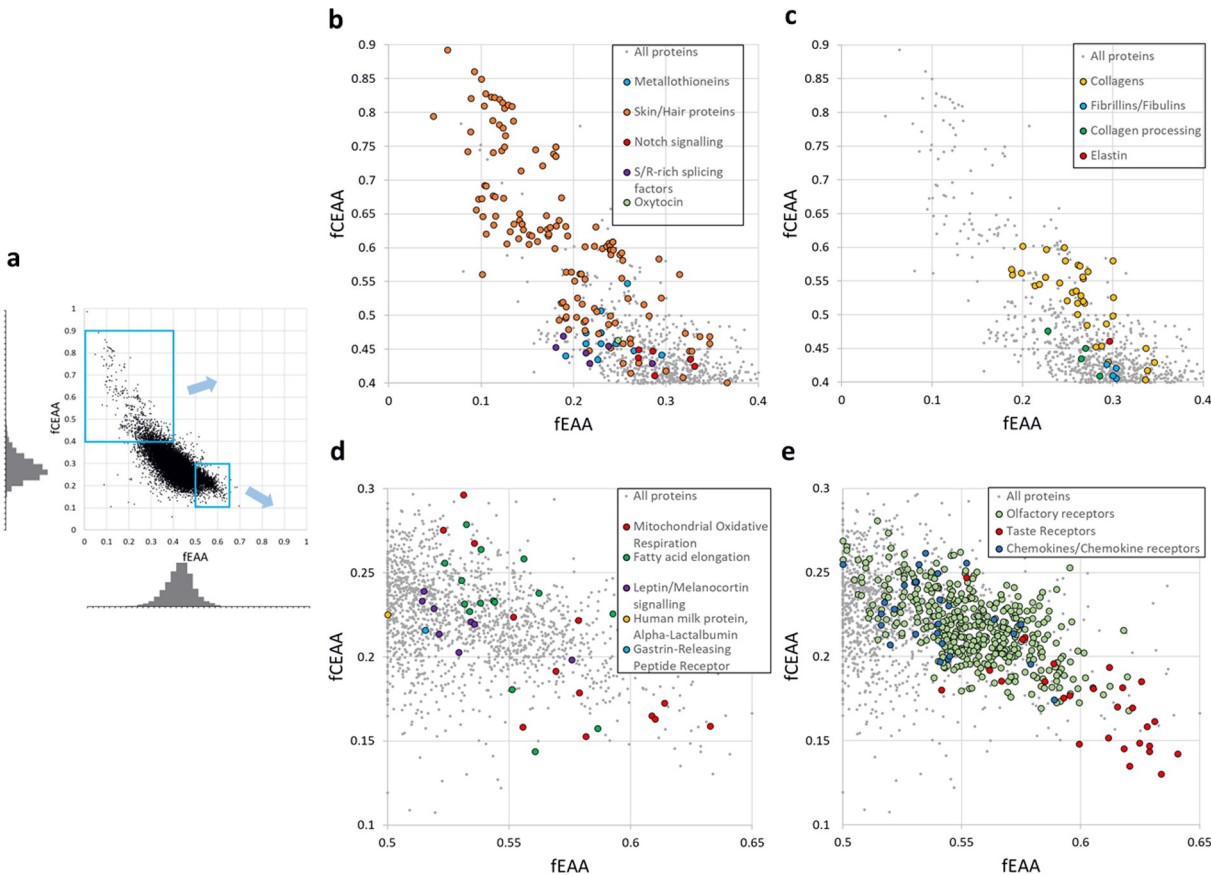

**Fig 4. EAA and CEAA over-representation in specific proteins and pathways reveals the consequences of, and response to, amino acid deprivation.** The composition of 20,397 human proteins was plotted, fEEA against fCEAA. (**a**) Tight distribution within a dense central cluster was clarified by the associated frequency histograms. Magnified outlier sectors of (**a**) contained protein families or functionalities with extremely high fCEAA (**b/c**) and fEAA (**d/e**) likely susceptible to reduced expression during nutritional insufficiency.

examined extreme outlier proteins for any insights into the biology and pathology of animal survival during malnourishment.

The extreme fCEAA outlier group primarily consisted of proteins with roles in the formation of connective tissue, skin, hair, and their maturation enzymes: collagens, elastin, keratin-associated proteins, late cornified envelope proteins, small cysteine, glycine and proline repeat-containing proteins, fibrillins, fibulins, lysyl oxidase, and latent-transforming growth factor beta-binding proteins (**Fig 4B and 4C**) [24, 25]. Gastrointestinal excretion and the production of skin and hair are the three principal routes of irretrievable amino acid loss from the body [26], so the relative paucity of EAAs in these proteins may be a resource conservation strategy. Furthermore, their extreme CEAA composition may act as a translational regulator for life processes that can be temporarily sacrificed or permanently downscaled to conserve energy and amino acid reserves. Anorexia nervosa can be accompanied by thinning hair, brittle nails, and deterioration in skin health [27] and, separately, periods of bodily stress or illness are frequently recorded as discontinuous nail growth in the form of Beau's lines or 'pitting'. Other proteins with high fCEAA that might be susceptible to the effects of dietary deficiency include all 14 metallothioneins involved in heavy metal-binding and oxidative stress responses, several members of the serine-/arginine-rich splicing factor family, oxytocin (a hormone involved in all aspects of reproduction and maternal resource investment from sexual arousal, to uterine contraction in labour, mother-offspring bonding, and milk production) and

multiple components of the Notch cell fate, differentiation, and injury repair signalling pathway (including DLL3, NOTCH4, and JAG2).

Extreme fEAA was observed in both leptin (LEP) and the melanocortin receptor proteins (MC1R-MC5R) (**Fig 4D**)–components of an established hypothalamic signalling pathway that responds to increased levels of adiposity/obesity by promoting satiety. This pathway is also linked to onset of puberty and stature [28]. GRPR (gastrin-releasing peptide receptor) also possesses high fEAA and similar appetite-suppressing role. We speculate that their conventional modes of signalling are augmented by 'hard-wired' protein synthesis inhibition during EAA deprivation–expression restriction of any of these proteins would act to increase appetite drive, potentially leading to recuperative ingestion of amino acids. Bitter taste receptors of the TAS2R subfamily have an extremely high EAA composition (**Fig 4E**) - with member TAS2R20 ranked 9th in the entire proteome. This contrasts with the proteome-average EAA composition of the umami and sweet taste receptors of the TAS1R subfamily. The TAS2R family fulfilled an important survival function in human prehistory by allowing detection and rejection of potentially toxic substances in foraged food. We make a speculative suggestion that bitter taste receptors have evolved a fragility of expression during dietary EAA deficiency. The resulting reduction in bitter taste may lower food discrimination or aversion, offering access to a greater range of foodstuffs potentially containing EAAs (histidine, tryptophan and valine salts are all bitter-tasting [29]). There is tentative evidence that prolonged anorexia nervosa blunts taste sensitivity [30]. Four other protein families have significant representation at the extremes of fEAA, including a group of fatty acid metabolism proteins (**Fig 4D**), 14 protein components of the mitochondrial electron transport chain complexes (**Fig 4D**), the large family of olfactory receptors (**Fig 4E**), and all 26 protein members of the CC-/CX- chemokine and chemokine receptor families that control chemotaxis and other immune cell functions (**Fig 4E**). Examination of the cell line data described earlier revealed globally reduced expression in the high fCEAA/high proliferation groups including the majority of the individual fCEAA outlier proteins discussed here. Highly proliferating lines showed specific reductions in expression of several fEAA outlier proteins, including most taste and olfactory receptors, some of the chemokines and their receptors, and slight decreases in expression for mitochondrial proteins such as MT-ND4, MT-ND5, and MT-ATP8.

*E. coli*, *S. cerevisiae*, and *A. thaliana* do not have compromised amino acid biosynthesis pathways requiring external EAA provision. However, they do possess the intrinsic 'constraints' defined in our earlier MLR findings as negative coefficient amino acids. We hypothesised that proteins at MLR-defined compositional extremes might also have been subject to evolutionary selection. A score was calculated for each protein based on combined negative coefficients (MLR-) and combined positive coefficients (MLR+) according to amino acid composition. Proteins with low MLR- values and high MLR+ values (expected to be highly expressed) were analysed via The Gene Ontology Resource [31] using Panther [32] to identify significant enrichment for specific biological processes (gene ontologies: GO). Next, for each significant GO term, MLR- and MLR+ scores were collated from the full set of proteins catalogued with those terms, and compared to the whole proteome using a two-tailed z-test to determine if these groupings had significantly different mean amino acid compositions (**Methods**). For *E. coli*, we observed significant enrichment within the less negative MLR-, more positive MLR+ protein sector for 251 proteins designated under the '*response to abiotic stimulus*' GO term (MLR- $p = 2.6 \times 10^{-5}$, MLR+ $p = 1.1 \times 10^{-6}$). The two least negative MLR- proteins in the entire *E. coli* proteome fall within this environment-detection category: acid shock protein (asr) and cold shock-like protein (cspC). Also significant were 116 proteins under the '*translation*' GO term (MLR- $p = 1.1 \times 10^{-18}$, MLR+ $p = 9.8 \times 10^{-61}$). This form of assessment is indirectly related to protein sequence and thus subject to bias from the presence of multiple

paralogs. Reanalysis of the paralog-rich *translation* term, collapsing multiple paralogs (e.g., ribosomal proteins, elongation factors) to a single averaged archetype, still yielded significance (MLR- p = 0.002, MLR+ p = $5.5 \times 10^{-8}$). *Translation*-related GO terms also presented statistically significant MLR score biases in *S. cerevisiae*, *A. thaliana* (root), and *H. sapiens* (liver). Likewise, environmental detection-related GO terms *response to high light intensity*, *water deprivation*, and *cold acclimation* (*A. thaliana*); *detoxification*, and *response to stress* (*H. sapiens*), all showed significant MLR score biases (**S2 File**).

### Amino acid composition and disease

In humans, gene-environment (GxE) interactions modifying disease risk and phenotypic expressivity may be encountered by proteins with extreme fEAA/fCEAA composition. Malnourishment in early life is currently experienced by 1 in 5 of the world's population, affecting stature, intellectual ability, future fertility, and risk of chronic illnesses–with the WHO reporting 145 million children with stunted height in 2020 [33, 34]. In the first approach to examine potential EAA/CEAA deficiency influences on disease risk, the online DisGeNET catalogue of genes associated with 8,383 diseases [35] was queried to identify extreme fEAA/fCEAA composition proteins which also had robust aetiopathological roles supported by at least 10 distinct disease annotations (**S2 File**). Proteins linked to cancers (e.g., the tumour suppressor, CDKN2A), CNS disorders (e.g., the myelin constituent, PMP22), and developmental disorders (e.g., skeleton and tooth development protein, SLC10A7) were represented at fCEAA and fEAA extremes. For fCEAA, there were many connective tissue disorders (due to the collagen protein family), as well as several proteins linked to miscarriage (COL5A1, IGFBP6, LGALS3); for fEAA, proteins were linked to immunological disorders, primarily due to the chemokine family and their receptors. These findings highlight potential dietary components to pathologies or treatments.

In a second, multigenic approach, five conditions (cancer, male infertility, female infertility, tooth abnormality, obesity) and one phenotype (stature/height) were chosen as established phenotypic indicators of malnourishment or, in the case of cancer, selected because of a pathology defined by aberrant proliferation. Risk proteins for each disorder were compiled from the literature or public databases and two-tailed Z-tests performed to determine if risk protein lists showed mean fCEAA or fEAA values significantly deviating from the entire proteome (**S1 File**). Significant findings were observed for cancer and stature. Established cancer proteins (n = 723, from the COSMIC Cancer Gene Census [36]) showed a highly significant under-representation of EAA (p = $0.00 \times 10^{0}$) and compensatory increase in fNEAA (p = $2.23 \times 10^{-31}$). Stature genes (n = 116, manually curated from literature) showed an increase in fCEAA (p = $1.02 \times 10^{-6}$) and a decrease in fEAA (p = $1.02 \times 10^{-05}$) although significance was largely driven by 8 members of the collagen family.

## Discussion

We have established both nutritionally governed and inherent mechanisms by which a protein's amino acid composition can influence its expression.

The profile of CEAA inhibitory effects on expression during baseline and proliferative cellular conditions offers the first rigorous molecular definition of this historically underexplored nutritional class. One consequence of our findings is that the proliferative state of laboratory cell lines (largely unreported in publication methods) may be a confounding factor for experimental replication of functional or expression studies - for half of all proteins. Expression of proliferation biomarker MKI67 may be a useful benchmark for such studies. By contrast, only modest consequences were observed for EAA-enriched proteins. Determining the true scale of

dietary EAA influences on protein expression *in vitro* and *in vivo* will require experimentation with amino acid-deficient culture media/feeds.

The remarkable second finding that individual amino acids affect protein expression in a largely conserved manner across species appears to be a consequence of the intrinsic amino acid properties of size, structural complexity, and codon allocation. It is presumed that these effects exert their influence at the ribosome during translation. We observed that proteins participating in translation and in species-specific environmental stress responses were significantly under-represented in amino acids with negative influence on expression and, conversely, over-represented in those with positive influence. We suggest this selective drive has ensured that survival-enhancing proteins can be rapidly, robustly, and highly expressed, even in challenging cellular and environmental conditions. Human and animal proteins at the extremes of EAA/CEAA composition may also have evolved as an advantageous strategy to survive nutritional scarcity. As well as the described effects on hair/nail/skin production and food-seeking behaviours, the modulation of collagen protein expression may be a key 'epigenetic' response to nutritional status in development: determining the limits of body size and appropriate maternal resource allocation - and aligning future metabolic demand (proportional to body scale) with anticipated environmental resource availability.

The findings presented here offer public health programs the prospect of quantitative protein biomarkers of clinical and sub-clinical amino acid dietary insufficiency. Additionally, they inform current clinical interventions based on amino acid supplementation and restriction. There are established benefits to amino acid supplementation (primarily the CEAAs glutamine or arginine) for patients undergoing ulcer treatment or post-surgery wound healing [37–40], potentially promoting connective tissue proliferation/regeneration. Supplementation with other CEAAs such as cysteine, or negative MLR coefficient amino acids such as serine, isoleucine, and tryptophan are now also worthy of investigation. By contrast, restricting amino acids in diet is an emerging concept in cancer treatment, capitalising on the specific demands of tumour cells. Our earlier findings on proliferation demands suggested that this 'hallmark' [41] would manifest as reduced fCEAA in cancer risk proteins. In fact, cancer proteins exhibited an extraordinary under-representation of EAA, suggesting that restricted essential amino acid supply to the tumour microenvironment may be a major determinant of protein expression, genotype-phenotype correlation, and clonal selection in cancer [42]. In tumours, expression of high fEAA proteins involved in mitochondrial oxidative respiration and chemokine function may also be compromised. This would be consistent with the Warburg effect [41] which describes the metabolic shift within tumours from oxidative respiration to glycolysis, and it may also contribute to the extensive chemokine/receptor-mediated interactions between tumour cells, stromal cells and macrophages [43]. Multiple amino acids have been trialled in restriction studies [44], mostly on the basis of gross abundance, so the detailed findings reported here may guide future dietary protocols in cancer treatment.

## Methods

### Data import and basic amino acid frequency analysis

From Uniprot.org, one representative human protein sequence per gene (totalling 20,397) was downloaded from the Reference Human proteome (ID: UP000005640) in FASTA format. Microsoft Excel text analysis formulas were applied to calculate the total amino residues, the frequency of each individual amino acid, and the relative proportions of the EAA/CEAA/NEAA nutritional classes present within each protein (**S1 File**). For example, a total of 34 EAA amino acids within a protein of 299 residues generates a fEAA of 0.11. Frequencies of amino acids or amino acid classes were used to remove the confounder of protein length differences.

Similar processes were carried out for the *Escherichia coli* (UP000635675), *Saccharomyces cerevisiae* (UP000002311), and *Arabidopsis thaliana* (UP000006548) proteomes.

## Protein expression correlation with amino acid classification

Protein expression data were imported as simple .txt files into Excel from publicly available datasets hosted in PaxDB. The human organ dataset IDs listed below represent freezes of multiple 'integrated' studies and so may change over time as more studies are incorporated. (Human tissues: Liver, #id: 11; Brain, #id: 180; Testis, #id: 1368841919; Heart, #id: 920111065; Pancreas, #id: 123; Ovary, #id: 249943175; Kidney, #id: 336; Lung, #id: 1177848913. Human cell lines: H441, #id: 2978442565; H1792, #id: 2702723926; H358, #id: 4263836592; H23, #id: 3320206519; A549b, #id: 878737823; A549a, #id: 3312331274; Cd9, #id: 2072474002; Hek293, #id: 1717546768; HepG2, #id: 4012390276; RKO, #id: 3395876293; Mcf7, #id: 4257247611; Hela, #id: 1648883758 & #id: 1603518140; Jurkat, #id: 1349617245; U2os, #id: 183123173 & #id: 931479677; K562, #id: 3816746548; Lncap, #id: 2209482342; Gamg, #id: 4023575820; A460, #id: 1189814538; B cell, #id: 2030745893; Nk, #id: 1493102864; Cd4, #id: 2689127103; H2122, #id: 1856553797; H727, #id: 928179904; Sklu1, #id: 1450716477. Yeast (*S. cerevisiae*), #id: 1329501331; Thale cress (*A. thaliana*), #id: 3612633737; Bacteria (*E. coli*), #id: 3836200197.). Species-specific protein identifiers in the expression data were converted into universal UniProt or UniPARC identifiers using the VLOOKUP command accessing imported conversion tables, allowing correlation with the amino acid/amino acid class frequencies of each protein.

## Moving median/average expression analysis

Nutritional amino acid class frequency was ranked and plotted against the moving median liver protein expression level (**Fig 1A**, periodicity of 469 = 5% of total). Multiple linear regression (see below) model score for each protein was plotted against liver expression value (log scale) and an Excel moving average trendline applied (periodicity of 100 proteins) (**Fig 3A and 3B and 3D**).

## Tissue and cell line plots of changing EAA/CEAA/NEAA proportions across expression levels

For a tissue or cell line, both the numerical expression level and the fEAA, fCEAA, and fNEAA for each protein were separately converted into ranks as a form of normalisation. For a protein ranked $n^{th}$ in expression, two figures were calculated for each nutritional amino acid class: the average proportion of that class from the lowest- to the $n^{th}$-ranked protein, and the average proportion of that class from the $n^{th}$- to the highest-ranked expression. The average of these two numbers was plotted for the $n^{th}$ protein (**Figs 1B–1G and S1 and S1 File**). This method produces smoothed plots of trends in relative amino acid class representation as a function of ranked protein expression level.

For Fig 1H–1I, protein expression values for each of 26 cell lines (see above) were rescaled to a value between 0–1 across 11,214 proteins for which expression was detectable in at least one cell line (proteins with no expression were excluded). Cell lines were stratified into three groups based on inferred proliferation rate (low proliferation rate, 11 lines; medium, 8 lines; high, 7 lines) as determined by the rescaled expression level of the proliferation marker MKI67 (Ki67). Proteins were additionally subdivided into 5 equal nutritional amino acid class frequency ranges (fEAA/fCEAA 1–5, with 5 having the greatest representation of that amino acid class). Total protein expression levels were calculated for each of the resulting 15 groups and plotted.

## Multiple linear regression and models

Multiple linear regression (MLR) in the Excel 'Data Analysis' ToolPak add-in was used to identify intrinsic parameters or individual amino acids with significant correlation to protein expression level, and their respective coefficients (**S1 File**). Statistically significant ($p<0.05$) MLR coefficients were normalised across organs and species in **Fig 2**. MLR findings allowed the construction of models which generated relative numerical expression predictions for each protein based on coefficients and amino acid frequencies.

## Statistical tests of gene ontology and disease candidate lists

fEAA and fCEAA figures for all proteins categorised within disease or phenotype subgroups were collated and means and variances calculated. These means were compared to the mean of the entire proteome. Analysis of the entire proteome showed statistically significant deviations in kurtosis and skewness for fEAA and fCEAA. However, the very large population size and small Lilliefors D effect size values of 0.043 (fEAA) and 0.078 (fCEAA) justified treating the distributions as effectively normal for the purpose of z-tests (**S1 File**).

Proteins located at the extremes of amino acid composition, as determined by the MLR +/- approach, were assessed for enrichment of specific gene ontology (GO) terms (Panther, The Gene Ontology Resource, https://geneontology.org/). For significantly enriched terms, all proteins categorised within that term were collated and assessed for amino acid composition. Two-tailed z-tests were applied to compare the MLR means in the entire proteome with proteins in gene ontology term groups.

Full monogenic and multigenic disease lists and statistical analyses relating to nutritional class composition are found in **S1 File**. Statistical analysis of enriched GO terms in MLR- and MLR+ data are found in **S2 File**.

## Supporting information

**S1 Fig. The relative proportions of the three nutritional amino acid classes change with protein expression level and proliferation rate (extended data from Fig 1B–1G).** A smoothing procedure (see **Methods** and **S1 File**) was applied to visualise trends in relative, ranked amino acid class proportion when plotted against ranked protein expression level for 8 human tissues and two samples of a lung cancer cell line, A549 (data from PaxDB). As described in the main text, tissues can be placed in three groups (**a**, **b**, and **c**) based on the profile of EAA/CEAA/NEAA composition across the range of expression levels. A549 differences (**d**) most likely represent amino acid constraint effects brought about by different proliferation rates. Graph **e** shows the same liver data as in **a** but with randomised expression level as a control. (TIF)

**S1 File. Multiple datasets and analyses comprising; amino acid composition calculator, human proteome AA composition, 25 selenocysteine-containing proteins, testing populations for normal distribution, calculating and smoothing relative proportions of EAA, CEAA, and NEAA in proteins as a function of expression level, multiple linear regression (MLR) analyses across organs, species, and amino acid parameters, extreme fEAA- & fCEAA-associated diseases, and multigenic disease statistics.** DOI: 10.15129/220e3e34-588b-4a94-93eb-8bd73cd2bf3e. (XLSX)

**S2 File. Statistics of the over-represented gene ontologies (GOs) in proteins with extremes of MLR coefficients.** DOI: 10.15129/220e3e34-588b-4a94-93eb-8bd73cd2bf3e. (XLSX)

## Author Contributions

**Conceptualization:** Benjamin Simon Pickard.

**Formal analysis:** Reece Thompson, Benjamin Simon Pickard.

**Methodology:** Benjamin Simon Pickard.

**Supervision:** Benjamin Simon Pickard.

**Visualization:** Benjamin Simon Pickard.

**Writing – original draft:** Reece Thompson, Benjamin Simon Pickard.

**Writing – review & editing:** Benjamin Simon Pickard.

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
