## [Decision Letter · Decision Letter 0]

24 Aug 2023

PONE-D-23-09085The amino acid composition of a protein influences its expressionPLOS ONE

Dear Dr. Pickard,

Thank you for submitting your manuscript to PLOS ONE. After careful consideration, we feel that it has merit but does not fully meet PLOS ONE’s publication criteria as it currently stands. Therefore, we invite you to submit a revised version of the manuscript that addresses the points raised during the review process. In your revised manuscript please address as fully as possible the points raised by the two reviewers, especiallythe detailed critique of Reviewer 2.

We look forward to receiving your revised manuscript.

Kind regards,

Israel Silman

Academic Editor

PLOS ONE

Journal Requirements:

"R.T. was funded through The Robertson Trust Internship Scheme 2021"

"RT: Internship funding for Summer 2021 from the Robertson Trust, UK. https://www.therobertsontrust.org.uk/

6. Please upload a new copy of Figures 1, 2 and 3 as the detail is not clear. Please follow the link for more information: https://blogs.plos.org/plos/2019/06/looking-good-tips-for-creating-your-plos-figures-graphics/’ https://blogs.plos.org/plos/2019/06/looking-good-tips-for-creating-your-plos-figures-graphics/

Reviewers' comments:

Reviewer's Responses to Questions

**Comments to the Author**

1. Is the manuscript technically sound, and do the data support the conclusions?

Reviewer #1: Yes

Reviewer #2: No

2. Has the statistical analysis been performed appropriately and rigorously? 

Reviewer #1: Yes

Reviewer #2: No

3. Have the authors made all data underlying the findings in their manuscript fully available?

Reviewer #1: Yes

Reviewer #2: No

4. Is the manuscript presented in an intelligible fashion and written in standard English?

Reviewer #1: Yes

Reviewer #2: No

5. Review Comments to the Author

Reviewer #1: Thompson and Pikcard study how amino acid composition affect the translation rate. The work is good provides a new perspective to how we look resource optimisation in protein synthesis. My comments on the manuscript are the following. 

It has been now established that cognate tRNA concentration influences the codon translation rates. Therefore, it is important to rule out whether the relationship authors found between amino acid composition and protein expression level is a direct consequence of that. 

Authors should also perform an analysis that how expression level, and functionality of proteins are connected with amino acid composition in a cell line. For example, I suspect that the expression of essential proteins (e.g. ribosomal protein) should be expressed robustly under all conditions. 

Other comments:

(1) Abstract is too long. It looks like an introduction, it should be shortened. 

(2) In Fig. 1(A), why didn't author measure correlation between expression level and enrichment of EAA and other amino acids.

Reviewer #2: The amino acid composition of a protein influences its expression.

The authors examined the amino acid composition of all proteins in the human genome and ranked them based on the frequency of three nutritional classes: essential, non-essential and conditionally essential. Then, they correlated each frequency with the expression levels of the proteins based on publicly available Mass Spectrometry data. With this, the authors observed that human proteins with higher expression tend to be enriched in essential amino acids, while those enriched in conditionally essential amino acids, needed from diet during specific contexts, tend to have lower expression. The authors have made the same calculation using multiple linear regression in Arabidopsis, Saccharomyces and E. coli. The frequency of the categories of amino acids correlates with expression in different species as well. Additionally, the enrichment of fCEAA amino acids seems to correlate with the function of proteins as enzymes of biosynthetic pathways that are sensible to be disrupted during periods of innutrition, such as during Anorexia.

Overall, the manuscript's readability is hindered by its speculative tone, utilization of intricate terms, and omission of key methodological details (as highlighted below). For instance, in the abstract, the authors allude to unveiling two protein expression regulation "rules," which, without context, lack clear definition and comprehensibility. This impedes the grasp of their implications in areas like systems biology, evolution, experimental design, and public health, as asserted by the authors. Moreover, certain terminology employed throughout the paper fails to accurately understand the depicted results.

The authors have quantified the occurrence of three amino acid types in proteins with varying expression levels. While the amino acid classification aligns with its origin, the authors interpret findings using terms like "supply constrain effects" and "amino acid supply kinetics." This could be perplexing as the actual measurement of amino acid supply, demand, intake, or individual levels hasn't been undertaken across the examined tissues or cell types.

Significantly, the manuscript consistently assumes uniform amino acid demand and supply across all analyzed tissues. When comparing different datasets, it raises the question of whether the definition of essential amino acids remains consistent among diverse tissues. Additionally, should the demand for amino acids remain identical across distinct tissues? These inquiries hold potential to significantly influence the interpretation of the data presented within this paper.

Points to raise:

- The specific identification numbers of the datasets employed by the authors has not been explicitly mentioned within the paper. While the supplementary data is indicated as the source for this information, it appears that these details have not been included for the reviewers' reference.

- The quality of the figures provided may benefit from improvement, as they currently appear to be somewhat challenging to decipher. This could be attributed to difficulties in reading the labels, which in turn might hinder the overall interpretation process.

- The absence of X or Y labels in the plots from Figure 1, b to h, could potentially pose challenges in comprehending the figure's content. As reviewers, we have inferred that these axes may align with the ones seen in Figure 1a. However, considering that the data originates from various tissues and cell types, this could lead to questions regarding the normalization process. For instance, when a dataset contains fewer proteins within each category, how is this aspect addressed in both the analysis and the plotted axes?

- Is it possible that various datasets from the same tissue type are being utilized? If that's indeed the situation, it would be valuable to understand how the data is being normalized. These particulars, which are currently absent from both the methods section and the overall content, play a crucial role in ensuring a transparent and accurate interpretation of the data.

- In figures 1 a-h, the authors have presented visualizations depicting the relative amino acid composition in relation to protein expression, utilizing organ expression datasets. Nonetheless, there appears to be a lack of clarification regarding the number of datasets utilized for each cell type or organ. This leads to an unanswered query regarding the standard deviation of the data. Additionally, it would be insightful to understand the approach employed for statistical analysis to ensure the robustness of the results.

- In Fig1e and f, the authors point out distinct profiles exhibited by two expression datasets derived from the A549 lung cancer cell line. They suggest a hypothesis that variations in tissue culture protocols across the source laboratories might influence amino acid identity. Similarly, it would be valuable for the authors to consider incorporating diverse datasets from the same tissues highlighted in the paper. This could involve calculating the standard deviation of the expression profiles, thereby enhancing the foundation for the conclusions drawn.

- In page 5 of the manuscript, the authors write “At both extremes, we interpret the apparent fNEAA influence on expression to be merely the passive consequences of active fCEAA or fEAA constraint effects”. This sentence could benefit from further clarity, as the authors haven't explicitly outlined the specific constraint effects they refer to. Additionally, it would be helpful to understand the authors' precise definitions for terms such as "active" and "passive" consequences.

- In page 5, the authors state that “the high-level protein expression requires amino acid composition to be near the population means”. This statement presents a conceptual challenge due to its wording, implying a necessity for proteins to possess specific amino acid compositions in order to achieve elevated expression levels. This concept is intricate to grasp, given that amino acid composition influences various aspects like function, folding, and enzymatic activity. Moreover, it's important to consider that the protein populations within each dataset vary in expression values due to their diverse tissue origins. Consequently, the amino acid composition of highly expressed proteins can vary across tissues.

- On page 6, the authors suggest a possible interpretation for the lower expression levels of proteins, proposing a connection to the efficiency of translation for conditionally essential and essential amino acids. It's worth noting that the measurement of translation efficiency hasn't been undertaken within the datasets employed in this study. Furthermore, it's important to acknowledge that translation efficiency represents just one of several molecular mechanisms governing protein levels. Aspects like protein degradation, post-translational modifications, and subcellular localization also contribute significantly.

- On page 7, the authors have made certain deductions concerning the proliferation rate. It's important to note that this data is associated with a normalized expression level obtained from the established proliferation marker Ki67 (MKI67) within each cell line. However, the specific methodology used for this calculation hasn't been thoroughly elaborated upon in the main text. This detail is significant to mention, particularly since the authors draw conclusions suggesting a potential negative impact of conditionally essential amino acids that is further influenced by a rapid proliferation rate.

-

- Is there an understanding regarding the uniformity of the definition of an essential amino acid across various datasets being compared? Additionally, has consideration been given to the potential variations in amino acid demand among different tissues?

- The authors propose that the higher presence of conditionally essential amino acids in proteins with lower expression levels may imply an insufficient biosynthetic capacity for these amino acids. The authors initially described this class of amino acids as those necessitating dietary supplementation during development or periods of stress or illness. Consequently, attributing the causation of amino acid scarcity based solely on protein expression levels appears inconclusive, given the unknown specifics of supplementation in the studied cell types, the actual amino acid consumption by each organ, and the dietary patterns of individuals within the datasets. The authors' conclusions are speculative in nature. To support their argument, it could be beneficial for the authors to establish correlations between the observed correlations and the expression levels of enzymes within biosynthetic pathways for the same amino acids in the examined tissues.

- In the abstract, the authors propose that "homeostatic responses to malnutrition may result from the reductions in expression of extreme composition proteins participating in biological systems such as taste and food-seeking behavior..." However, the authors have yet to present substantial evidence exclusively linking this phenomenon to the essentiality of amino acids. Moreover, malnutrition can lead to dysfunctions in organ and cellular processes that may not necessarily be connected to amino acid nature. How do the authors distinguish between the impact of amino acid sequences on proteins and the potential influence of other biochemical processes disrupted by the stress of malnourishment?

- In figures 1h and 1i, the authors introduce the concept of amino acid supply kinetics to interpret the data. However, it's worth noting that the actual amino acid levels within the cells haven't been quantified; therefore, the interpretation should revolve solely around the amino acid composition of the proteins. Furthermore, the explanation provided for Fig1i poses some difficulty in comprehension, which in turn hinders the formation of clear conclusions: "Secondly, a largely proliferation independent, positive effect of increasing fEAA on protein expression levels was observed which switches to negative only for those proteins with the very highest fEAA (perhaps determined by the limits of EAA availability in media and its cellular uptake)."

- On page 10, the authors mentioned, "We observed a moving average expression spanning three orders of magnitude when plotting the liver MLR model value for each protein against its true liver expression." However, we are finding it challenging to fully comprehend the implications of this sentence.

- Could the authors please provide an explanation of the Akashi, Wagner, and Zhang models, as well as the Dufton score discussed on page 10 of the manuscript? Understanding what these models measure and the rationale behind their selection would greatly enrich the manuscript and its interpretation.

- On page 12 of the manuscript, the authors state: “we theorized that such proteins might have retained counterintuitively extreme compositions to sense and respond to environmental adversity”. Could the authors kindly rephrase this sentence for clarity?

- Figure 3 appears to illustrate the potential abundance of conditionally essential amino acids within enzymes participating in certain biosynthetic pathways, which could be influenced by dietary deficiencies. However, further clarification regarding the specific statistical analysis employed to arrive at these findings is not provided by the authors. Furthermore, the section discussing the methodology for "statistical tests of gene ontology and disease candidate list" in the methods could benefit from clearer explanation. For example, out of the total proteins in each biosynthetic term, how many are found enriched in conditionally essential amino acids?

- “The resulting reduction in bitter taste and smell acuity may lower food discrimination or aversion, offering access to a greater range of foodstuffs potentially containing EAAs…” The authors seem to be engaging in considerable speculation regarding the potential causes of a complex behavior, primarily based on the observed frequency of amino acid occurrence in human proteins.

- As the supplementary information has not been provided for reviewers' assessment, the section pertaining to "Amino Acid Composition and Disease" within the manuscript has not been accessible for review.

6. PLOS authors have the option to publish the peer review history of their article (what does this mean?). If published, this will include your full peer review and any attached files.

Reviewer #1: **Yes: **Ajeet Sharma

Reviewer #2: No

---

## [Author Response · Author response to Decision Letter 0]

12 Oct 2023

Response to reviewers' comments:

We thank the reviewers for their considered criticism of the submitted manuscript and believe the current version has benefitted considerably from their input.

Reviewer #1: Thompson and Pickard study how amino acid composition affect the translation rate. The work is good provides a new perspective to how we look resource optimisation in protein synthesis. My comments on the manuscript are the following. 

It has been now established that cognate tRNA concentration influences the codon translation rates. Therefore, it is important to rule out whether the relationship authors found between amino acid composition and protein expression level is a direct consequence of that. 

>We mention the influence of aminoacyl-tRNA synthesis on protein translation in the Introduction as we also agree it is an important and established factor in translation efficiency. Moreover, the second described form of amino acid effect in our manuscript (where individual amino acid compositions influence expression across all species) is further dissected into amino acid structure and amino acid codon property effects – the latter clearly implicates aminoacyl-tRNA choice and availability in the regulation of expression. However, the animal nutritional class amino acid effects on protein expression that we have described do not necessarily require or implicate a role for alternative tRNA concentrations.

Authors should also perform an analysis that how expression level, and functionality of proteins are connected with amino acid composition in a cell line. For example, I suspect that the expression of essential proteins (e.g. ribosomal protein) should be expressed robustly under all conditions. 

>We agree that this is a fundamental aspect to determine. We believe we have already demonstrated this in the manuscript in two ways. Firstly, the right-hand sides of Fig.1b-f show that the most highly expressed proteins in a cell (among which are ribosomal proteins, translation elongation factors, and nucleosomal proteins) all have amino acid class compositions that are similar to the whole proteome frequencies of EAA/CEAA/NEAA (EAA, 0.41; fCEAA, 0.28; fNEAA, 0.31) – in other words, they have evolved compositions that avoid the risks to expression of extreme amino acid composition. We have amended the main text to make this aspect clearer and annotated the figure and changed the legend to allow readers to appreciate this fact. Secondly, when looking at the influence of individual amino acid composition on protein expression across all species we observed that the ‘Translation’ gene ontology term (which includes ribosomal proteins) was significantly enriched in all studied species for amino acids promoting robust expression (Results Section: ‘protective biological systems at extremes of amino acid composition’).

Other comments:

(1) Abstract is too long. It looks like an introduction, it should be shortened. 

>We have simplified and shortened the Abstract and removed mention of the ‘Rules’ that Ref#2 found unclear.

(2) In Fig. 1(A), why didn't author measure correlation between expression level and enrichment of EAA and other amino acids.

> We do calculate correlation for the second interaction between individual amino acids and protein expression (Table 1) and acknowledge in the text that these correlations, albeit hugely statistically significant, are small in magnitude. Because the interactions between amino acids and expression are largely felt at the extremes of composition, a simple correlation obscures effect size in these regions. Therefore, we have not added a correlation calculation to Fig.1a.

Reviewer #2: The amino acid composition of a protein influences its expression.

The authors examined the amino acid composition of all proteins in the human genome and ranked them based on the frequency of three nutritional classes: essential, non-essential and conditionally essential. Then, they correlated each frequency with the expression levels of the proteins based on publicly available Mass Spectrometry data. With this, the authors observed that human proteins with higher expression tend to be enriched in essential amino acids, while those enriched in conditionally essential amino acids, needed from diet during specific contexts, tend to have lower expression. The authors have made the same calculation using multiple linear regression in Arabidopsis, Saccharomyces and E. coli. The frequency of the categories of amino acids correlates with expression in different species as well. Additionally, the enrichment of fCEAA amino acids seems to correlate with the function of proteins as enzymes of biosynthetic pathways that are sensible to be disrupted during periods of innutrition, such as during Anorexia.

Overall, the manuscript's readability is hindered by its speculative tone, utilization of intricate terms, and omission of key methodological details (as highlighted below). For instance, in the abstract, the authors allude to unveiling two protein expression regulation "rules," which, without context, lack clear definition and comprehensibility. This impedes the grasp of their implications in areas like systems biology, evolution, experimental design, and public health, as asserted by the authors. Moreover, certain terminology employed throughout the paper fails to accurately understand the depicted results.

>Throughout the MS, we have made significant amendments to the text that remove any overly intricate words/phrases in the description of the data and its interpretation for the sake of clarity. For example, we have removed all reference to the two ‘rules’ of amino acid effect on protein production, and merely describe them through their differing modes of action. We have also largely removed terms such as intrinsic/extrinsic/immutable/heterotrophic/prototrophic. More generally, we have simplified the descriptions of our observations to promote clarity and readability.

The authors have quantified the occurrence of three amino acid types in proteins with varying expression levels. While the amino acid classification aligns with its origin, the authors interpret findings using terms like "supply constrain effects" and "amino acid supply kinetics." This could be perplexing as the actual measurement of amino acid supply, demand, intake, or individual levels hasn't been undertaken across the examined tissues or cell types.

>We have rewritten the descriptions of our results to provide greater clarification of what we mean by ‘supply’ and ‘constraint’, and how they are just one explanation for the observations and that they are not the result of direct measurement. Phrases such as ‘supply kinetics’ have been removed to avoid confusion and to avoid the implication that specific kinetics investigations have been carried out.

Significantly, the manuscript consistently assumes uniform amino acid demand and supply across all analyzed tissues. When comparing different datasets, it raises the question of whether the definition of essential amino acids remains consistent among diverse tissues. Additionally, should the demand for amino acids remain identical across distinct tissues? These inquiries hold potential to significantly influence the interpretation of the data presented within this paper.

>We have already suggested in the original version that the different organ profiles ‘…reflect inherent organ features such as non-proteogenic amino acid use (gluconeogenesis), local proliferation rate, basal metabolic rate, or amino acid transport capacities’. However, we would argue that this does not necessarily indicate that the definitions of nutritional amino acid classes are wrong, as these were defined at the whole organism level through dietary restriction studies, and by our modern biochemical understanding of amino acid synthesis pathways. Additionally, the evidence from isolated A549 cells suggests that these different demands can indeed be ‘state-dependent’ (e.g., through proliferation rate) rather than fixed. We have amended the text to better address the reviewer’s thoughts on specific organ demands: ‘In this way, organ-specific categories of ‘essential’ and ‘conditionally essential’ amino acids may practically differ from the whole organism definition.’

Points to raise:

- The specific identification numbers of the datasets employed by the authors has not been explicitly mentioned within the paper. While the supplementary data is indicated as the source for this information, it appears that these details have not been included for the reviewers' reference.

>All PaxDB dataset identifiers have now been added to the Methods section.

- The quality of the figures provided may benefit from improvement, as they currently appear to be somewhat challenging to decipher. This could be attributed to difficulties in reading the labels, which in turn might hinder the overall interpretation process.

>All figures have been completely reworked, relabelled, rescaled and accompanied by complementary changes to legends. Image files are in high-resolution TIFF format. 

- The absence of X or Y labels in the plots from Figure 1, b to h, could potentially pose challenges in comprehending the figure's content. As reviewers, we have inferred that these axes may align with the ones seen in Figure 1a. However, considering that the data originates from various tissues and cell types, this could lead to questions regarding the normalization process. For instance, when a dataset contains fewer proteins within each category, how is this aspect addressed in both the analysis and the plotted axes?

>As above for improved figures. The images are based on (i) relative abundances of the three classes of amino acid within a protein (ii) ranked expression level. Both aspects allow direct comparison between independent datasets without any further normalisation. The total number of proteins with observable expression may vary between organs/studies, and X-axes have been stretched/compressed to compensate, but with the total number still remaining in the thousands, there is no impact on analysis/graphing. More explicit descriptions of this process have been made in the text and figures have improved labelling to get across these important points.

- Is it possible that various datasets from the same tissue type are being utilized? If that's indeed the situation, it would be valuable to understand how the data is being normalized. These particulars, which are currently absent from both the methods section and the overall content, play a crucial role in ensuring a transparent and accurate interpretation of the data.

>The Methods text has been improved to describe the rescaling/normalisation processes employed to allow comparisons and create figures. See below for a comprehensive description of the tissue sources.

- In figures 1 a-h, the authors have presented visualizations depicting the relative amino acid composition in relation to protein expression, utilizing organ expression datasets. Nonetheless, there appears to be a lack of clarification regarding the number of datasets utilized for each cell type or organ. This leads to an unanswered query regarding the standard deviation of the data. Additionally, it would be insightful to understand the approach employed for statistical analysis to ensure the robustness of the results.

>In most cases, the PaxDB datasets used were integrated at source from multiple studies and, hence, represent an intrinsically reliable sample coverage. The text has been amended to include this as well as the identifiers.

“Protein expression data were imported as simple .txt files into Excel from publicly available datasets hosted in PaxDB. The human organ dataset IDs listed below represent freezes of multiple ‘integrated’ studies and so may change over time as more studies are incorporated. (Human tissues: Liver, #id: 11; Brain, #id: 180; Testis, #id: 1368841919; Heart, #id: 920111065; Pancreas, #id: 123; Ovary, #id: 249943175; Kidney, #id: 336; Lung, #id: 1177848913. Human cell lines: H441, #id: 2978442565; H1792, #id: 2702723926; H358, #id: 4263836592; H23, #id: 3320206519; A549b, #id: 878737823; A549a, #id: 3312331274; Cd9, #id: 2072474002; Hek293, #id: 1717546768; HepG2, #id: 4012390276; RKO, #id: 3395876293; Mcf7, #id: 4257247611; Hela, #id: 1648883758 & #id: 1603518140; Jurkat, #id: 1349617245; U2os, #id: 183123173 & #id: 931479677; K562, #id: 3816746548; Lncap, #id: 2209482342; Gamg, #id: 4023575820; A460, #id: 1189814538; B cell, #id: 2030745893; Nk, #id: 1493102864; Cd4, #id: 2689127103; H2122, #id: 1856553797; H727, #id: 928179904; Sklu1, #id: 1450716477. Yeast (S. cerevisiae), #id: 1329501331; Thale cress (A. thaliana), #id: 3612633737; Bacteria (E. coli), #id: 3836200197.).”

- In Fig1e and f, the authors point out distinct profiles exhibited by two expression datasets derived from the A549 lung cancer cell line. They suggest a hypothesis that variations in tissue culture protocols across the source laboratories might influence amino acid identity. Similarly, it would be valuable for the authors to consider incorporating diverse datasets from the same tissues highlighted in the paper. This could involve calculating the standard deviation of the expression profiles, thereby enhancing the foundation for the conclusions drawn.

>The comment above addresses this point – the PaxDB datasets employed are integrated from multiple studies.

- In page 5 of the manuscript, the authors write “At both extremes, we interpret the apparent fNEAA influence on expression to be merely the passive consequences of active fCEAA or fEAA constraint effects”. This sentence could benefit from further clarity, as the authors haven't explicitly outlined the specific constraint effects they refer to. Additionally, it would be helpful to understand the authors' precise definitions for terms such as "active" and "passive" consequences.

>The text has been amended to better describe the observations, and our interpretations. ‘Active’ was relating to frequency changes that result from the nutritional constraints on expression, ‘passive’ relates to the fact that all frequencies have to add up to 1. Extremely low NEAA frequencies ‘could’ be postulated to explain reduced expression but the corresponding high EAA/CEAA frequencies provide a much more logical mechanistic explanation. In that example, EAA/CEAA changes are the active effect, and NEAA changes the passive consequence. 

- In page 5, the authors state that “the high-level protein expression requires amino acid composition to be near the population means”. This statement presents a conceptual challenge due to its wording, implying a necessity for proteins to possess specific amino acid compositions in order to achieve elevated expression levels. This concept is intricate to grasp, given that amino acid composition influences various aspects like function, folding, and enzymatic activity. Moreover, it's important to consider that the protein populations within each dataset vary in expression values due to their diverse tissue origins. Consequently, the amino acid composition of highly expressed proteins can vary across tissues.

>These are very interesting points which get to the heart of the findings. The key phenomenon uncovered is actually that some proteins show extreme deviation from the whole-proteome means of amino acid composition – whereas highly-expressed proteins tend not to. Accordingly, we have ‘flipped’ the emphasis in the description of these effects: highlighting the extremes of amino acid composition with their negative effects on expression, and concluding that highly expressed proteins have simply avoided this sequence bias. Specific new text: ‘Of note, these observations suggest that the majority of proteins have arrived at an evolutionary balance between the selection of appropriate amino acids (to meet requirements of structure/functionality), and the selection of amino acid compositions that avoid supply constraints on expression. Aside from hydrophobic amino acids typically being EAAs, there is relatively little alignment between nutritional amino acid classes and the functional amino acid classifications (polar/acidic/basic, etc.) suggesting a degree of flexibility in amino acid selection.’ To address the final point, we are interrogating only the limitations on proteins that are detectably expressed in a given tissue/cell. Those that are not expressed will have an expression of 0 and will not be analysed because there may not be any corresponding mRNA expression. This is why mass spectrometry proteomics rarely has significantly more than 10,000 quantified proteins (from the possible ~20,000). Within each tissue, we are analysing the expression effects of amino acid composition on only detectable proteins. Where relevant, this has been added to the Methods text.

- On page 6, the authors suggest a possible interpretation for the lower expression levels of proteins, proposing a connection to the efficiency of translation for conditionally essential and essential amino acids. It's worth noting that the measurement of translation efficiency hasn't been undertaken within the datasets employed in this study. Furthermore, it's important to acknowledge that translation efficiency represents just one of several molecular mechanisms governing protein levels. Aspects like protein degradation, post-translational modifications, and subcellular localization also contribute significantly.

>We do not claim that constraints on amino acid supply can explain all aspects of translation regulation (as acknowledged in a reworked version of the first paragraph of the Introduction), but we don’t believe that the listed alternatives provide a logical explanation for our various findings relating to expression changes when proteins are stratified on the basis of long-established dietary differences in amino acid origin.

- On page 7, the authors have made certain deductions concerning the proliferation rate. It's important to note that this data is associated with a normalized expression level obtained from the established proliferation marker Ki67 (MKI67) within each cell line. However, the specific methodology used for this calculation hasn't been thoroughly elaborated upon in the main text. This detail is significant to mention, particularly since the authors draw conclusions suggesting a potential negative impact of conditionally essential amino acids that is further influenced by a rapid proliferation rate.

>As detailed above, we acknowledge that the Methods section required modification and addition to allow the manuscript to stand alone from the supplementary sections. We hope these changes make the data transformations clear.

- Is there an understanding regarding the uniformity of the definition of an essential amino acid across various datasets being compared? Additionally, has consideration been given to the potential variations in amino acid demand among different tissues?

>We have addressed this point above.

- The authors propose that the higher presence of conditionally essential amino acids in proteins with lower expression levels may imply an insufficient biosynthetic capacity for these amino acids. The authors initially described this class of amino acids as those necessitating dietary supplementation during development or periods of stress or illness. Consequently, attributing the causation of amino acid scarcity based solely on protein expression levels appears inconclusive, given the unknown specifics of supplementation in the studied cell types, the actual amino acid consumption by each organ, and the dietary patterns of individuals within the datasets. The authors' conclusions are speculative in nature. To support their argument, it could be beneficial for the authors to establish correlations between the observed correlations and the expression levels of enzymes within biosynthetic pathways for the same amino acids in the examined tissues.

> Cell culture media is extremely rich in amino acids, particularly so since the transition from EM to DMEM media formulations. To our knowledge, the individuals that provided samples for all of the independent proteomics datasets used in this study were all adequately nourished. So, to see a consistent profile of CEAA inhibition that is different in profile from EAA effects, and also exacerbated by proliferation, suggests a ‘cell autonomous’ (internal biosynthesis rate) rather than environmental cause. Rate-limited production of these amino acids is a convincing explanation. The idea that the underlying synthetic enzymes are likely to show evidence for upregulation in these stress conditions is a very intriguing suggestion from the reviewer. We examined this, comparing high- and low- proliferating cells looking for synthetic enzyme or transporter expression differences. Aside from MTRR (methionine synthase reductase), no obvious correlations with proliferation rate were seen. This enzyme contributes to methionine regeneration from homocysteine, but this is equally likely to feed methylation reaction substrate generation rather than proteogenesis.

- In the abstract, the authors propose that "homeostatic responses to malnutrition may result from the reductions in expression of extreme composition proteins participating in biological systems such as taste and food-seeking behavior..." However, the authors have yet to present substantial evidence exclusively linking this phenomenon to the essentiality of amino acids. Moreover, malnutrition can lead to dysfunctions in organ and cellular processes that may not necessarily be connected to amino acid nature. How do the authors distinguish between the impact of amino acid sequences on proteins and the potential influence of other biochemical processes disrupted by the stress of malnourishment?

>We have reworked the text to emphasise the speculative nature of these proposed explanations and have removed the weaker speculation relating to olfactory receptors. However, we believe that some reference to outlier protein function is merited. The discovery of an apparently strong stabilising selection pressure on their extreme protein composition demands some attempt at hypothesis formation, if only to enable future experimental testing.

- In figures 1h and 1i, the authors introduce the concept of amino acid supply kinetics to interpret the data. However, it's worth noting that the actual amino acid levels within the cells haven't been quantified; therefore, the interpretation should revolve solely around the amino acid composition of the proteins. Furthermore, the explanation provided for Fig1i poses some difficulty in comprehension, which in turn hinders the formation of clear conclusions: "Secondly, a largely proliferation independent, positive effect of increasing fEAA on protein expression levels was observed which switches to negative only for those proteins with the very highest fEAA (perhaps determined by the limits of EAA availability in media and its cellular uptake)."

>As indicated above, we have made significant changes to the main text to clarify our descriptions of the data.

- On page 10, the authors mentioned, "We observed a moving average expression spanning three orders of magnitude when plotting the liver MLR model value for each protein against its true liver expression." However, we are finding it challenging to fully comprehend the implications of this sentence.

>The correlations between the model-predicted expression and the ‘real’, experimentally-derived expression for each protein were always going to be complex, imperfect and hard to visualise, particularly when a log scale is required by the significant dynamic range of protein expression. Therefore, we applied a ‘moving average’ approach to illustrate the trend. This calculates the average predicted and actual expression levels across a range (the ‘window’) of proteins and then moves this incrementally through the population of proteins as sorted by predicted expression. We have rephrased the text to say: ‘The pan-species nature of these amino acid effects was effectively demonstrated by using testing the ability of the E. coli MLR model from Table 1 and Supplementary File 1 to predict trends in actual global human protein expression (Fig. 2). Firstly, a moving average trendline between the original human liver MLR model value for each protein against that protein’s true liver expression was plotted (Fig 2a). From lowest to highest ‘window’ of prediction, the corresponding average protein expression increased by three orders of magnitude, as might be expected given this was the source data for the MLR model. However, applying the bacterial MLR model to human liver expression still successfully predicted a range of average liver expression spanning two orders of magnitude (Fig 2b).’

- Could the authors please provide an explanation of the Akashi, Wagner, and Zhang models, as well as the Dufton score discussed on page 10 of the manuscript? Understanding what these models measure and the rationale behind their selection would greatly enrich the manuscript and its interpretation.

>We were keen to identify the underlying causes for the individual amino acid effects on protein expression and were guided by the literature towards three parameters of amino acids: how they are coded, how easily they are made, and their ultimate structure. The main text has been amended and simplified to clarify the rationale for these choices.

- On page 12 of the manuscript, the authors state: “we theorized that such proteins might have retained counterintuitively extreme compositions to sense and respond to environmental adversity”. Could the authors kindly rephrase this sentence for clarity?

>The opening of this section has been largely rewritten to aid comprehension.

- Figure 3 appears to illustrate the potential abundance of conditionally essential amino acids within enzymes participating in certain biosynthetic pathways, which could be influenced by dietary deficiencies. However, further clarification regarding the specific statistical analysis employed to arrive at these findings is not provided by the authors. Furthermore, the section discussing the methodology for "statistical tests of gene ontology and disease candidate list" in the methods could benefit from clearer explanation. For example, out of the total proteins in each biosynthetic term, how many are found enriched in conditionally essential amino acids?

>The statistical Methods section has been significantly upgraded and the corresponding results sections in the main text expanded to explain rationale.

- “The resulting reduction in bitter taste and smell acuity may lower food discrimination or aversion, offering access to a greater range of foodstuffs potentially containing EAAs…” The authors seem to be engaging in considerable speculation regarding the potential causes of a complex behavior, primarily based on the observed frequency of amino acid occurrence in human proteins.

>As detailed above, we have reworked the text to emphasise the speculative nature of the text but feel that some attempt to hypothesise the extreme amino acid compositions and effects on protein expression is warranted – and testable in the future. The olfactory section has been de-emphasised.

- As the supplementary information has not been provided for reviewers' assessment, the section pertaining to "Amino Acid Composition and Disease" within the manuscript has not been accessible for review.

>The Supporting Information File data was submitted for review. A DOI has been generated as described in the comments to the editors.

---

## [Decision Letter · Decision Letter 1]

6 Nov 2023

The amino acid composition of a protein influences its expression

PONE-D-23-09085R1

Dear Dr. Pickard,

We’re pleased to inform you that your manuscript has been judged scientifically suitable for publication and will be formally accepted for publication once it meets all outstanding technical requirements.

Kind regards,

Israel Silman

Academic Editor

PLOS ONE

Additional Editor Comments (optional):

Reviewers' comments:

Reviewer's Responses to Questions

**Comments to the Author**

1. If the authors have adequately addressed your comments raised in a previous round of review and you feel that this manuscript is now acceptable for publication, you may indicate that here to bypass the “Comments to the Author” section, enter your conflict of interest statement in the “Confidential to Editor” section, and submit your "Accept" recommendation.

Reviewer #1: All comments have been addressed

2. Is the manuscript technically sound, and do the data support the conclusions?

Reviewer #1: Yes

3. Has the statistical analysis been performed appropriately and rigorously? 

Reviewer #1: Yes

4. Have the authors made all data underlying the findings in their manuscript fully available?

Reviewer #1: Yes

5. Is the manuscript presented in an intelligible fashion and written in standard English?

Reviewer #1: Yes

6. Review Comments to the Author

Reviewer #1: The authors have addressed all the questions/concerns. The quality of manuscript has now improved substantially.

I have no further comments on this manuscript. The manuscript can be accepted in its current form.

7. PLOS authors have the option to publish the peer review history of their article (what does this mean?). If published, this will include your full peer review and any attached files.

Reviewer #1: No

---

## [Editor Report · Acceptance letter]

15 Dec 2023

PONE-D-23-09085R1 

PLOS ONE

Dear Dr. Pickard, 

I'm pleased to inform you that your manuscript has been deemed suitable for publication in PLOS ONE. Congratulations! Your manuscript is now being handed over to our production team.

Kind regards, 

on behalf of

Prof. Israel Silman 

Academic Editor

PLOS ONE